# Detection and Phylogenetic Analysis of Caprine Arthritis Encephalitis Virus Using TaqMan-based qPCR in Eastern China

**DOI:** 10.3390/vetsci11030138

**Published:** 2024-03-21

**Authors:** Yutong Tian, Hailong Zhang, Yan Zhang, Xinya Zhang, Zhilei Guan, Junjie Zhang, Yafeng Qiu, Beibei Li, Ke Liu, Zongjie Li, Donghua Shao, Peng Li, Zhiyong Ma, Jianchao Wei

**Affiliations:** 1College of Animal Science, Yangtze University, Jingzhou 434025, China; sweetieupup@163.com; 2Shanghai Veterinary Research Institute, Chinese Academy of Agricultural Sciences, Shanghai 200241, China; zhanghailong1997@163.com (H.Z.); 18672550235@163.com (Y.Z.); 13770510390@163.com (X.Z.); 17639342748@163.com (Z.G.); 17317271403@163.com (J.Z.); yafengq@shvri.ac.cn (Y.Q.); lbb@shvri.ac.cn (B.L.); liuke@shvri.ac.cn (K.L.); lizongjie@shvri.ac.cn (Z.L.); shaodonghua@shvri.ac.cn (D.S.); 3College of Veterinary Medicine, Hebei Agricultural University, Baoding 071001, China; 4College of Veterinary Medicine, Nanjing Agricultural University, Nanjing 210095, China

**Keywords:** caprine arthritis encephalitis virus, TaqMan RT-qPCR, goat, phylogenetic trees, detecting method, infectious diseases

## Abstract

**Simple Summary:**

Caprine arthritis encephalitis is an infectious disease caused by the caprine arthritis encephalitis virus (CAEV). This virus can impair the quality of goat’s milk, lamb, and other goat products. Most goats are latently infected with this virus and become long-term carriers, so the development of a rapid and accurate diagnostic method is necessary. Therefore, we constructed a TaqMan real-time quantitative polymerase chain reaction (RT-qPCR) assay and used it to detect CAEV in sheep in eastern China. We found that the prevalence of CAEV infection in sheep in eastern China was 0.77%. This method yields positive results when testing positive references and negative results when testing negative controls or other viruses. The consistency of these results was confirmed through repeating the experiment multiple times. It demonstrated excellent sensitivity, specificity, and reproducibility and holds great promise for its potential application in clinical and field samples in the future.

**Abstract:**

Caprine arthritis encephalitis is an infectious disease caused by the caprine arthritis encephalitis virus that infects goats, sheep, and other small ruminants. An outbreak of CAEV could be extremely harmful to the goat farming industry and could cause severe economic losses. We designed specific primers and probes for the gag gene and established a TaqMan real-time quantitative polymerase chain reaction assay. This method’s correlation coefficient (R^2^) was >0.999, and the sensitivity of the assay to the plasmid-carried partial gag gene was approximately 10 copies/µL, 1000 times higher than that of conventional PCR. No specific fluorescence was detected for other sheep viruses. Using this method, we tested 776 asymptomatic sheep blood samples and 4 neurodegenerative sheep brain samples from six farms in eastern China, and the positivity rate was 0.77% (6/780). The gag gene was partially sequenced in the three positive samples and compared with the sequences from other representative strains in GenBank. The results revealed that all three strains belonged to the B1 subtype and were most closely related to the strains from Shanxi and Gansu, previously isolated in China, with their homology ranging from 97.7% to 98.9%. These results suggest that the designed RT-qPCR assay can be used to detect subclinical CAEV in sheep and that the virus is still present in eastern China.

## 1. Introduction

Caprine arthritis encephalitis is a chronic progressive infectious disease caused by the caprine arthritis encephalitis virus (CAEV). CAEV belongs to the family Retroviridae, subfamily Lentiviridae, genus Small Ruminant Lentivirus (SRLV) [1]. SRLV, which includes caprine arthritis encephalitis virus (CAEV) and Maedi visna virus (MVV), causes a classical type of chronic progressive disease that is characterized by a long eclipse period, a slow progression of the disease, and persistent infection. MVV is widely spread across Europe, including in Switzerland, Italy, Iceland, and elsewhere, and only rarely reported in China, with all cases having been located in Sichuan and Xinjiang provinces; hence, this study focused on CAEV, which is more prevalent in China and covers a wider area [2,3].

CAEV is a small, single-stranded enveloped RNA virus with a genome length of 9.2 kb, containing three genes encoding structural proteins—gag (containing a group-specific antigen), env (including an envelope protein), and pol (polypeptide enzyme)—two genes encoding regulatory proteins—vif and tat—and one gene encoding an accessory protein—rev [4]. Gag and pol are the most conserved regions [5]. During SRLV infection, the majority of animals produce the protein encoded by the gag gene early in the infection process, making the RNA transcribed from the gag gene a commonly utilized diagnostic marker for SRLV [6]. In studies of strains isolated from goats infected with MVV and CAEV, it has been observed that, in the context of mixed infections, the genomes of CAEV and MVV undergo recombination, thereby complicating the classification of CAEV [7,8,9,10]. The first reported occurrence of caprine arthritis encephalitis traces back to Switzerland in 1974. Subsequently, in 1982, CAEV was isolated from the swollen joint mucus of afflicted goats [11]. In the early 1980s, China imported Shannon dairy goats from the United Kingdom on three separate occasions, which resulted in CAEV being introduced to China [12]. At present, CAEV infection exists in China; its prevalence remains at 16~19% [13]. Based on the conserved sequences of gag (>1 kb) and pol, SRLV can be classified into five genotypes: A to E. Type A is further classified into 15 subtypes (A1 to A15), and type B is classified into 3 subtypes (B1 to B3). Previous studies have found that subtypes B to D and some subtypes of A (A1 to A6, A9, and A11 to A13) can infect both goats and sheep [14,15,16,17]. The predominant strains circulating in China belong to subtype B1.

CAEV transmission routes encompass both horizontal and vertical transmission [18]. Diseased sheep excrete the virus through nasal secretions, feces, milk, etc. The virus infects healthy sheep through contaminated forage or drinking water and milk. While gastrointestinal transmission serves as the primary route of CAEV transmission, respiratory transmission and transplacental transmission are also documented pathways [19,20]. The prevalence of CAEV is influenced by factors such as the sheep’s breed, flock size, living conditions, medical status, introductions, and the trade of live sheep [21]. Goats, sheep, and other small ruminants are susceptible to CAEV infection; goats are the most susceptible small ruminants [20]. Clinically, there are four main types of CAEV infection: encephalomyelitis (neurologic), polyarthritis, interstitial pneumonia, and mastitis [22]. Clinical symptoms are observed in only 20 to 30 percent of affected goats; in sheep and other small ruminants, CAEV only presents with subclinical symptoms. The primary clinical manifestations of a CAEV infection in lambs typically include acute encephalomyelitis. In adult goats, common clinical presentations involve polyarthritis, interstitial pneumonia, and sclerosing mastitis [23]. The disease has a worldwide distribution and a high prevalence in many countries, along with a long incubation period. Infected goats carry the virus throughout their lives, as there is no specific treatment for the disease, eventually leading to death. There are no effective control measures and no drugs or commercial vaccines against CAEV. Efforts to control the infection are now relying on biosecurity management.

CAEV damages the central nervous system (CNS) of sheep, and if the disease was to cause an epidemic, it would bring huge losses to the sheep farming industry and also pose a great threat to public health and socio-economic development [24]. The clinical manifestations of CAEV lack specificity during their initial diagnosis, posing a challenge in differentiating this virus from other viruses causing encephalitis in sheep. As there is no effective treatment available for this disease, the key lies in early detection, and stringent control measures are the most effective method in managing CAEV. A variety of laboratory diagnostic techniques have been established for the detection of CAEV, including the agarose gel diffusion (AGID) test, enzyme-linked immunosorbent assay (ELISA), nucleic acid in situ hybridization, nucleic acid probes, nested PCR, and polymerase chain reaction (PCR) [25,26]. The AGID test and ELISA are the diagnostic methods recommended by the World Organization for Animal Health (WOAH); however, the AGID test and ELISA cannot effectively detect low antibody levels during the incubation period of the viral infection. In China, only virus isolation, PCR, the agar gel immunodiffusion test, and the c-ELISA for CAEV are recognized in the industry standards issued by policymakers in the agricultural and rural sectors. Since the first successful isolation of poliovirus from cells by Ebders in 1949, virus isolation followed by molecular confirmation has been the gold standard for the determination of viruses. However, due to the stringent experimental conditions required for this technique, the isolation cost is high, the experimental period is long and complicated, and highly skilled operators are required. For highly pathogenic viruses, a highly sophisticated laboratory is necessary, otherwise there will be a greater risk to biosafety. Numerous studies have shown that, compared with the ELISA, Western blotting, and PCR, AGID has poor sensitivity, and its results are often false negatives [27,28,29,30]. Theoretically, qPCR methods are applicable to SRLV-infected animals in all stages of infection, including the early stage of the infection and the incubation period. The fluorescence RT-qPCR technique has gained widespread acceptance for its use in the diagnosis of various viruses due to its sensitivity, rapidity, and specificity. Among the various reagents used for RT-qPCR, the TaqMan probe method is superior to the SYBR dye method due to its improved specificity and sensitivity. Fluorescence quantitative PCR has good specificity and accuracy, simple experimental steps, and a short reaction time; therefore, it is currently the most widely used virus detection method [31].

## 2. Materials and Methods

### 2.1. Sample Collection

Sheep samples (*n* = 634; 630 blood samples and 4 brain samples) were collected between July and September 2021 from three sheep farms in Shanghai. In July 2022, we collected 146 goat blood samples from three goat farms in Jiangsu, China. Of all the clinical samples, only four sheep showed neurological symptoms, and the farm provided the brains of these sheep. The blood samples were randomly collected from asymptomatic animals. All samples were delivered to the lab in Drikold and then stored at −80 °C until use (Appendix A). All experimental procedures involving animals were performed in adherence to the Guidelines for the Keeping and Use of Laboratory Animals, and approval for this study was obtained from the Animal Ethics Committee of the Shanghai Institute of Veterinary Medicine, Chinese Academy of Agricultural Sciences. This field study did not involve endangered or protected species. No specific permissions were required for the collection of these samples because the samples were collected from public areas or nonprotected areas. We received approval from the farm owners for data sampling and the publication of the data.

### 2.2. Primers and Probes

To construct primers and probes, the nucleotide sequences of the whole genomes of CAEV were retrieved from the GenBank database and compared, using the CLUSTAL V algorithm in the sequence comparison software MegAlign (MegAlign 5.00, DNASTAR Inc., Madison, WI, USA). The nucleotide sequence of the gag gene in the CAEV reference strain was obtained from GenBank (no. GU120138.1) and compared with the nucleotide sequences of other CAEV strains (NC001463.1 JX982691.1, MN233151.1, AY900630.1, KT749881.1, KT214469.1, AJ900630.1). TaqMan probes and primers tailored to the conserved regions of the genomes of the viruses were prepared using Snapgene 4.1.9 version software. To determine whether the selected target regions were suitable for a quantitative real-time polymerase chain reaction (qRT-PCR), the primers were synthesized and comprehensively evaluated. Table 1 contains comprehensive details of the nucleotide sequences of the primer pairs and probes specific to CAEV and the properties of the amplicons that were produced.

### 2.3. RNA Extraction and Reverse Transcription

The clinical samples were processed in two ways. The blood samples were diluted with phosphate-buffered saline (PBS) and directly used for RNA extraction; however, the brain samples were added to PBS and homogenized using a tissue grinding bead beater. Then, the homogenate was used for RNA extraction. All samples were extracted using a nucleic acid extraction kit (MANMAN Biotechnology Co., Shanghai, China) and following the manufacturer’s instructions. A portion of the extracted RNA was stored at −80 °C, and another portion was used for reverse transcription into cDNA using the Evo M-MLV Reverse Transcription Kit (Accurate Biotechnology Co., Ltd., Changsha, China), and this cDNA was stored at −20 °C.

### 2.4. The TaqMan RT-qPCR Method

The RT-qPCR reactions consisted of 10 μL 2 × TaqMan mixture (Accurate Biotechnology Co., Ltd.), 250 nM (0.5 μL) forward primer, 250 nM (0.5 μL) reverse primer, 250 nM (0.5 μL) ROX-labeled probe, 6.5 µL RNase-free water, and 2.0 µL cDNA. Before testing on the CFX96 real-time PCR detection system (Bio-Rad Co., Ltd., Washington, DC, USA), the standardized cycle conditions for CAEV were set as follows: 98 °C for 1 min; 35 cycles of 95 °C for 15 s; and 53 °C for 30 s.

### 2.5. Standard Curve Generation

The synthesized CAEV standard plasmid was quantified using the NanoDrop 1000 spectrophotometer (Thermo Fisher Scientific, Waltham, MA, USA), and the copy number of the plasmid was calculated using the following formula: [copies/μL = 6 × 10^23^ × ssRNA (ng/μL) × 10^−9^ ÷ molecular weight (g/mol)]. A standard plasmid of CAEV (10^9^ copies/μL~10^1^ copies/μL) was serially diluted 10-fold to prepare a standard to assess and compare the sensitivity of our RT-qPCR with that of the conventional RT-PCR assay. The standard curve for CAEV detection was created using the 10-fold serial dilution standards as templates.

### 2.6. Specificity and Sensitivity Analysis of TaqMan RT-qPCR Assay

The specificity of the method was determined using inactivated forms of the following viruses in specific experiments: CAEV, PPRV, BTV, FMDV, PRV, and GPV. For the sensitivity assays, the constructed CAEV clone plasmid was diluted in a 10-fold gradient for the comparison of the TaqMan RT-PCR method to the conventional PCR method. In the conventional PCR method, we used industry-standard primers (relative to the Chinese agricultural industry) (Table 1). Inactivated foot-and-mouth disease virus (FMDV, KR073010.1), peste des petits ruminants virus (PPRV, OL310687.1), goatpox virus (GPV, MG458414.1), pseudorabies virus (PRV, U38548.1), and bluetongue virus (BTV, OP185812.1) were kindly supplied by the Shanghai Branch of the Chinese Centre for Animal Hygiene and Epidemiology. Standard strains of caprine arthritis encephalitis and its plasmids were synthesized by Huajin Biological Company; they just contained the sequences of the conserved regions mentioned in Section 2.2 (length: 372 bp).

### 2.7. Repeatability Determination of TaqMan RT-qPCR Assay

An established method was used to test the intra-assay and inter-assay repeatability of the TaqMan RT-qPCR assay. The Ct value was obtained by amplification, and the coefficient of variation (CV%) was calculated to evaluate the repeatability of this method (Table 2).

### 2.8. Detecting of Clinical Samples

The optimized qPCR assay was used to detect and analyze cDNA prepared from 780 samples collected from six farms in eastern China. The farms in Jiangsu provided goat samples, and the Shanghai farms provided sheep samples. The sizes of the farms and the quantities of samples provided are illustrated in Table 3.

### 2.9. Construction of Phylogenetic Tree

The PCR products of 3 positive samples that could be detected by conventional PCR were analyzed by electrophoresis using 2% agarose gel under UV light. The positive bands were sliced, and the gel was purified using the Gel Extraction Kit (Omega, Doraville, GA, USA) and sequenced by Sangon bio-company (Shanghai, China). The primer sets we used for sequencing are just the primer sets we used in the conventional PCR method, which were referred to in Section 2.6. They were named SH2023, JS2023-1, and JS2023-2. Referring to the studies of other scholars, we selected 28 representative sequences. These sequences were classified into defined groups and applied to construct phylogenetic trees several times. The sequences of the 28 representative CAEV species were downloaded from GenBank and edited to obtain the corresponding segments using Editseq version 7.1.0 software (DNASTAR Inc., Madison, WI, USA; Lole et al., 1999), based on the sequencing primers. Along with the sequences of the Shanghai and Jiangsu isolates SH2023, JS2023-1, and JS2023-2, multiple sequence alignments were performed using the Clustal V method, and a phylogenetic tree was constructed using DNASTAR version 7.0 software (DNASTAR Inc., Madison, WI, USA; Lole et al., 1999).

## 3. Results

### 3.1. Designing Specific Primers and Probes

From the whole genome sequencing of CAEV, primers and probes were designed to target the gag gene. Their detailed sequences and specific regions are shown in Table 1 and Figure 1, respectively. Using Primer-BLAST (NCBI), the primers and probes were analyzed; they exhibited high conservation and specificity and could be used for the specific detection of CAEV.

### 3.2. Preparation of the Standard Curve of Caprine Arthritis Encephalitis Virus

For the preparation of the standard curve of caprine arthritis encephalitis virus, the PUC-57 vector was used to create the CAEV strain plasmid. To obtain a reliable calibration curve in the real-time RT-PCR assay, we employed a linear range of 10^9^~10^1^ copies/μL copy (10-fold). The linear regression equation for the standard curve of CAEV was y = −2.646 ∗ x + 37.10 (R^2^ = 0.999) (Figure 2). The correlation coefficient (R2) of the virus was 0.999, demonstrating the great reliability of this technique for virus detection. The standard curves were utilized to determine the number of genomic RNA copies of CAEV in the clinical samples used in the subsequent real-time RT-qPCR assays.

### 3.3. The Specificity of the RT-qPCR Assay

In addition to PPRV, sheep are also commonly infected with FMDV, GPV, PRV, and BTV. Thus, to establish the specificity of this assay, RNA of PPRV, FMDV, GPV, PRV, and BTV was used (Figure 3). The ROX fluorescent signal was discovered only when the template was CAEV. The outcomes demonstrated the significant effectiveness of this method in distinguishing CAEV from other viruses. This approach showed great specificity for CAEV, as no positive signals were detected when other viruses were used as templates.

### 3.4. The Sensitivity of the RT-qPCR Assay

The TaqMan RT-qPCR method and the conventional PCR method were utilized to detect and evaluate the differences in the sensitivity of the two methods using 10-fold serial dilutions of the CAEV standard as a template. This method evidently had a high degree of sensitivity because it was 10^3^ times more sensitive than the conventional PCR method in identifying CAEV (Figure 2).

### 3.5. Intra-Assay and Inter-Assay Reproducibility of the RT-qPCR Assay

To test the reproducibility of this method, the prepared 10-fold continuous dilution of the standard was subjected to intra-assay and inter-assay repeatability tests. Their respective coefficients of variation for CAEV were less than 1% and 1.4%. Therefore, the method exhibited good reproducibility (Table 2).

### 3.6. Detection of Samples from Farms in Eastern China

Six farms in eastern China provided 780 clinical samples, including 634 sheep samples (630 blood samples and 4 brain samples) from Shanghai farms and 146 goat blood samples from Jiangsu farms. The application of the TaqMan RT-qPCR assay to analyze the 780 clinical samples revealed a positive presence of CAEV in six samples, with cycle threshold values (ct) ≤ 35. Notably, only three of the samples among the six detected using the TaqMan RT-qPCR tested positive for CAEV when the conventional PCR method was employed. The TaqMan RT-qPCR showed that 0.77% (6/780) of the samples were positive for CAEV, and 99.23% (774/780) were negative. The six positive animals came from three farms—one Jiangsu farm (JS-1) and two Shanghai farms (SH-1 and SH-3)—and they all showed no clinical signs of disease (Table 3).

### 3.7. Phylogenetic Analyses of Caprine Arthritis Encephalitis Virus

We submitted the three sequences isolated in this study to GenBank and obtained the following sequence numbers: JS-2023-1 (no. PP382771), JS-2023-2 (no. PP382772), and SH-2023 (no. PP382773). As previously reported, there are no epidemiologic surveys of the East China region (the provinces Jiangsu, Zhejiang, Shanghai, and Anhui). In order to examine the dominant circulating genotypes and emerging strains in eastern China, we compared the 3 sequences isolated in this study with 28 sequences we downloaded from GenBank and constructed a phylogenetic tree. Based on the phylogenetic trees and extant findings, phylogenetic analyses were conducted. These phylogenetic analyses revealed that the three sequences isolated in this study all belong to the B1 subtype, so the B1 subtype remains dominant in China. The genomic homology of the Shanghai isolate and the two isolates from Jiangsu was 82.1% and 87.1%, respectively, while the homology between the two Jiangsu isolates was 98.9%. Compared with the sequences available in GenBank, SH-2023 exhibited the highest genomic similarity with the isolates from Gansu and Shanxi, both at 97.74%; JS-2023-1 had the highest genomic similarity with the isolates from Gansu and Guizhou, both at 98.77%, and so did JS-2023-2, but with a slightly higher genomic similarity of 97.80%. To conclude, all three isolates belong to the B1 subtype (Figure 4).

## 4. Discussion

CAEV infections are prevalent in dairy goats all over the world, and a higher incidence of CAEV has been noted in many industrialized nations. In this study, a high-sensitivity and high-specificity assay based on TaqMan real-time PCR amplification was developed for detecting CAEV. Using the method established in this study, we tested samples from flocks in eastern China (Shanghai and Jiangsu) and found relatively low CAEV positivity in eastern China (around 0.77%), but the virus has not been eliminated completely. This result was lower than expected, and we attribute this deviation to the guidelines outlined in the document titled “Diagnostic Techniques for CAEV” (NY/T 3465-2019), issued in China in 2019. This document establishes the technical requirements for the clinical diagnosis of CAEV, protocols for the isolation and identification of pathogens, methods for detecting pathogenic nucleic acids, and guidelines for serological tests.

In China, CAEV has been a concern for the caprine farming industry since the first domestic infection caused by CAEV in 1988. Epidemiological investigations of CAEV in eastern China trace back to 1995, at which time CAEV was not found to be circulating in goat flocks in that region. Since then, there have been no publicly published detection reports on CAEV in eastern China. China put forward the “Eighth Five-Year Plan” national scientific and technological research project in the 1990s. The prevention and treatment of CAEV was an important topic in this plan. During this period, Chinese scientists conducted an extensive examination of the virus, covering a total of 11,449 goats from 11 different breeds across 15 provinces in China. They employed clinical and serological methods, revealing varying degrees of CAEV’s prevalence in most provinces: Gansu (5.14%), Shaanxi (9.25%), Yunnan (0.52%), Guizhou (7.45%), Henan (1.08%), Shandong (6.16%), Hainan (4.18%), Liaoning (1.80%), Sichuan (0.42%), Xinjiang (0.18%), and Heilongjiang (0.11%) [32]. ELISA methods are suitable for large-scale sample testing but are usually only used for antibodies in serum, and ELISAs require multiple steps to be carries out, resulting in longer experimental times. Despite the ongoing efforts of scientists in the post-Eighth Five-Year Plan era, research on the prevention and control of CAEV is still lacking, and the results concerning this topic are not fully conclusive. In 2012, a Chinese academic conducted tests on 308 clinical samples from Tianjin and Shaanxi, utilizing mononuclear peripheral blood cells from goat blood as target cells, resulting in a positive rate of 7.8% [33]. In 2013, some Chinese researchers conducted an investigation on 308 goat samples collected between 1993 and 2011 in the Shanxi and Tianjin provinces of China. They employed a TaqMan RT-qPCR assay, revealing a positive rate of 7.8% [34]. Compared to previous methods, the TaqMan RT-qPCR assay developed in this study demonstrates enhanced stability, reduced error rates in its results, and heightened specificity compared to previously established RT-qPCR assays, particularly in distinguishing CAEV from other prevalent infectious diseases in sheep. In 2018, a Chinese academic conducted tests on 165 goat samples from the Xinjiang region, employing both molecular and serological assays, resulting in a positivity rate of 2.4% [35]. In the same year, Chinese scientists tested 2083 goat samples from 11 provinces in China, revealing a positivity rate of 0.38% [36]. A comparison of the results of previous studies with the results of this study suggests that the prevalence of CAEV is higher in the western region of China than in the east and higher in the northern region than in the south. We speculate that this may be because the practice of sheep farming is more developed in the northwestern region. Meanwhile, in East China regions like Shanghai and Jiangsu, where disease prevention and control measures, comprehensive management capabilities, and public safety regulations are more stringent, sheep farms enforce strict regulations on trade. This, we hypothesize, may have contributed to the lower positive rate observed in our study.

We tried to isolate the strains that tested positive, but it was difficult to do so. Hence, we resorted to sequencing to obtain the sequence of their gag gene, and compared the three isolates with the sequences already uploaded to the NCBI to obtain a phylogenetic tree. From the tree, we inferred that subtype B has spread very widely, appearing in countries on many different continents, including China, the United States, Canada, Poland, Switzerland, France, Brazil, Italy, and Spain. Subtype B1 is still the most prevalent subtype in China. Compared to the northwestern and southwestern regions of China, the practice of sheep farming in eastern regions is less developed, and, thus, sheep farmers in this region are more likely to neglect and underestimate the presence of infectious diseases in their flocks. As a chronic infectious disease, CAEV is difficult for breeders to detect in its first instance, meaning that regular screening is necessary to prevent this disease from causing serious economic losses to the sheep farming industry [37]. CAEV is more common in goats. However, there is evidence of the direct transmission of the SRLV subtype B1 from goats to sheep, and our study also reaffirms the validity of the transmission of subtype B1 in sheep. These results suggest that it is best to raise goats and sheep separately and avoid mixing them [10].

CAEV eradication programs are hindered by late seroconversion, as observed in goats, and by the absence of detectable antibodies in infected animals, which delays diagnosis and promotes the dissemination of the disease [38]. Therefore, pathogenetic detection methods play an important role in CAEV eradication programs. The establishment of this optimized TaqMan RT-qPCR assay provides a technical means for the clinical detection of CAEV. Not only can specific pathogens be rapidly detected in the early stages of epidemics and appropriate response strategies be formulated in a timely manner, but the epidemiological and developmental trends of diseases can also be monitored, and scientific research can be promoted. We hope that this assay will contribute to the sheep farming industry and provide long-lasting benefits.

## 5. Conclusions

Our results showed that the TaqMan RT-qPCR method designed in this study is a molecular assay for CAEV that has good specificity, sensitivity, and stability, and that it can rapidly and accurately identify CAEV in clinical tests. Through applying this method, we obtained a positivity rate of 0.77% for sheep in eastern China. Due to its high similarity to the gag gene of the strain prevalent in other regions of China, we concluded that the cross-region transmission of CAEV in sheep flocks in China is a problem that needs to be monitored by the relevant authorities.

## Figures and Tables

**Figure 1 vetsci-11-00138-f001:**
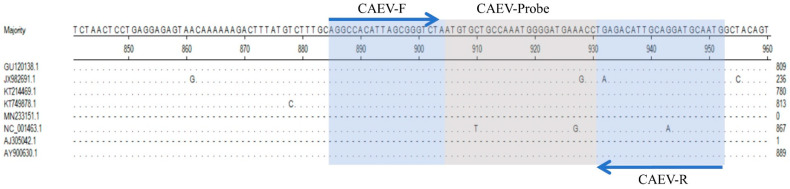
Comparison of the gag gene of different strains of caprine arthritis encephalitis virus (CAEV) and the positions of the primers and TaqMan probes in the viral genome. The GenBank accession number is shown in parentheses. The dots (∙) indicate identical bases.

**Figure 2 vetsci-11-00138-f002:**
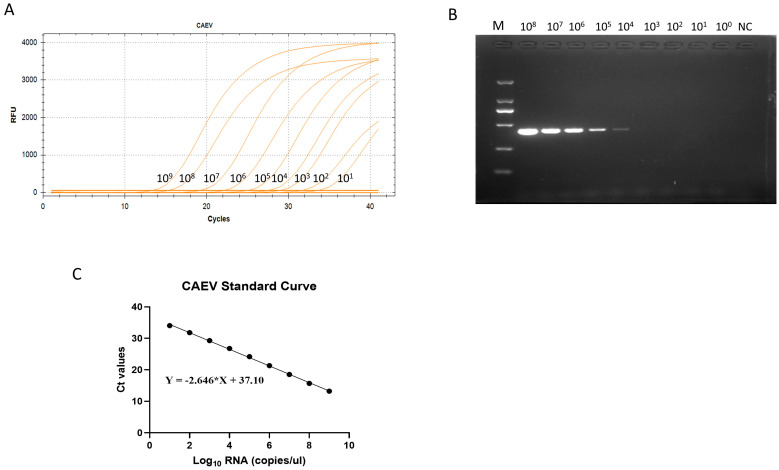
(**A**) The amplification curves showing the sensitivity of the TaqMan-based real-time quantitative polymerase chain reaction (RT-qPCR) in detecting caprine arthritis encephalitis virus (CAEV). (**B**) The sensitivity of the conventional PCR toward CAEV (10^9^ copies/μL~10^0^ copies/μL). (**C**) The standard curve of the TaqMan RT-qPCR.

**Figure 3 vetsci-11-00138-f003:**
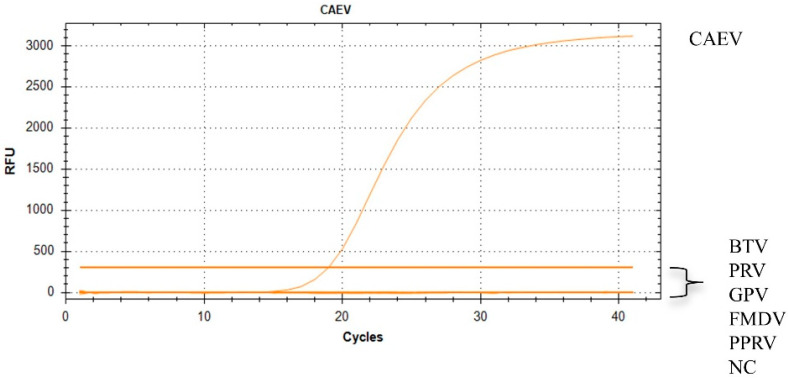
Specific amplification curves for CAEV. A specific fluorescent curve was observed in the TaqMan RT-qPCR assay for CAEV with RNA mixtures. ROX fluorescent signals specific for CAEV were detected only when CAEV isolates were used as templates. No ROX signal was observed in the samples containing other viruses.

**Figure 4 vetsci-11-00138-f004:**
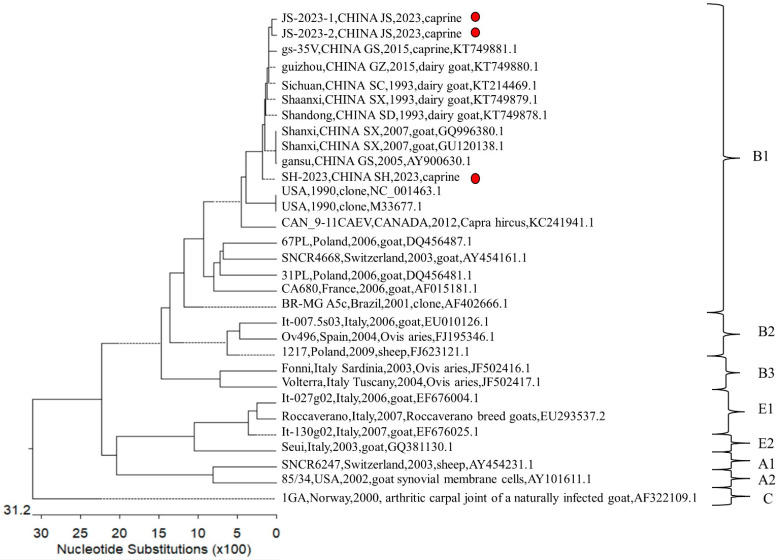
Phylogenetic tree of caprine arthritis encephalitis virus (CAEV) based on the partial gag gene (about 571 bp). The three strains isolated in this study—JS-2023-1 (no. PP382771), JS-2023-2 (no. PP382772), and SH-2023 (no. PP382773)—are marked by red circles. The other 28 representative sequences were downloaded from GenBank and edited to obtain the corresponding segments using Editseq version 7.1.0 software (DNASTAR Inc., Madison, WI, USA; Lole et al., 1999) based on the sequencing primers used. Multiple sequence alignments were performed using the Clustal V method, and a phylogenetic analysis of the recombinants was performed using MegAlign version 7.1.0 software (DNASTAR Inc., Madison, WI, USA; Lole et al., 1999).

**Table 1 vetsci-11-00138-t001:** Oligonucleotide primers and fluorogenic probes used in TaqMan RT-qPCR assay and primers used in conventional PCR assay.

Virus	Location	Primer/Probe	Sequences (5′-3′)	Length (bp)
CAEV	gag	Primer-CAEV-F	ATGTCTTTGCAGGCCACATT	68
		Primer-CAEV-R	TGCAATGTCTCAGGTTTCATCC	
		Probe-CAEV	^ROX-^CCATTTGGCAGCACATTAGACCCGC^-BHQ−1^	
CAEV	gag	CAEV-F	AACTGGAAAGCAGTAGAC	571
		CAEV-R	TACACTAGCTTGTTGCAC	

**Table 2 vetsci-11-00138-t002:** Reproducibility of the TaqMan RT-qPCR assay, evaluated using ssRNA standards of CAEV.

Virus Standards	Copy Number	Intra-Assay		Inter-Assay	
		CT Value (Mean ± SD)	CV (%)	CT Value (Mean ± SD)	CV (%)
CAEV	10^9^	13.16 ± 0.06	0.3636	14.86 ± 0.19	0.6621
	10^8^	15.55 ± 0.14	0.7356	18.15 ± 0.07	0.2131
	10^7^	18.44 ± 0.18	0.6902	21.87 ± 0.54	1.3212
	10^6^	21.30 ± 0.05	0.1232	25.13 ± 0.20	0.4617
	10^5^	24.23 ± 0.06	0.1856	28.29 ± 0.05	0.1212
	10^4^	26.56 ± 0.23	0.6158	31.93 ± 0.49	0.8793
	10^3^	29.38 ± 0.25	0.6095	34.34 ± 0.26	0.7571
	10^2^	31.70 ± 0.42	0.9550	36.18 ± 0.47	1.2991
	10^1^	34.34 ± 0.11	0.2615	38.56 ± 0.28	0.7261
	10^0^	NaN	NaN	NaN	NaN
	NC	NaN	NaN	NaN	NaN

**Table 3 vetsci-11-00138-t003:** Detection of CAEV in clinical samples from goats and sheep using TaqMan RT-qPCR assay between 2021 and 2022.

Location	Farm	Farm Size	Type of Sample	Sample Collection	Clinical Symptoms	NO. of Samples	NO. of Positive Samples
Jiangsu	JS-1	500–1000	Goat Blood	Jugular	No	34	2 (JS-2023-1, JS-2023-2)
	JS-2	>2000	Goat Blood		No	77	0
	JS-3	>2000	Goat Blood		No	35	0
Shanghai	SH-1	>2000	Sheep Blood	Jugular	No	352	3 (SH-2023)
			Sheep Brain	Dissection	Yes	4	0
	SH-2	500–1000	Sheep Blood	Jugular	No	150	0
	SH-3	500–1000	Sheep Blood	Jugular	No	128	1
Total						780	6

## Data Availability

The datasets used and/or analyzed during the current study are available from the corresponding author on reasonable request.

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
