# Peer review of "Detection and Phylogenetic Analysis of Caprine Arthritis Encephalitis Virus Using TaqMan-based qPCR in Eastern China"

_vetsci, 2024, doi:10.3390/vetsci11030138_

Round 1
Reviewer 1 Report
Comments and Suggestions for Authors
The manuscript requires extensive correction before being acceptable- Also the Genbank accession numbers for the sequences must be available: OTHERWISE, THE MANUSCRIPT CAN NOT BE ACCEPTED.
Besides, at particular points, I have marked these comments:
-Line 21: The phrase “ This method demonstrated excellent 22 sensitivity, specificity, and reproducibility “ needs a demonstration, even in the abstract. For a given test Sensitivity is usually defined as the probability of a positive test result, when an individual is truly positive, while specificity is the probability of a negative result when an individual is a real negative. Both must be compared somehow with the diagnosis obtained in a different way. Lines 29-31 later explain that sensitivity is tested “in vitro” by using known concentrations of a plasmid carrying the GAG gene. Please clarify this “in vitro” validation for both sensitivity and specificity.
-Line 33. The term “genomes” usually refers to complete genomes. For a single gene (like GAG), it is more adequate to write “the sequence of the GAG gene” (for instance)
-Line 41 and successive: I strongly recommend dividing the first very long paragraph in the Introduction into, at least, three shorter paragraphs: very different ideas are exposed, and they surely deserve
-Line 49: why using the words “containing” and “including”?
-Line 53: This gene or this protein? Or the RNA transcribed from this gene? Please write a more clear sentence.
-Lines 59-61. This sentence requires a reference: in case the virus has been demonstrated to have reached China in this way, there must be a publication somewhere reporting this fact.
-Lines 61 and 62: again, a reference is needed to support the present low prevalence.
-Line 66: please re-write the phrase “The main prevalent strains in China are all subtype B1” and “The routes of CAEV transmission include: horizontal and vertical transmission”
-Line 70: do you mean transplacental transmission?
-Line 73: I suggest “...; goats are the most …”
-Line 76. I suggest “affected sheep”
-Line 77. Please re-write this phrase.
-Line 89-92. Please re-write this phrase on the different procedures. The present form is a bit confusing.
-Line 99: Please consider a different word for the “level” of the operator (for instance “highly skilled operators”?)
-Line 108: the number of brain samples and of blood samples is required. Also, goat samples are obtained in sheep farms in the line 110: is this correct?
-Line 119-121: the word compared appears twice. Please re-write this phrase.
-Line 123-124: either you explain how the primers were evaluated or you eliminate this phrase on the evaluation.
-Table 1: Usually the numbering is used to explain the position of the primers. This Table seems to explain the size of the product. Besides, the term (V = A/G/C). is not necessary since no V is visible in the sequences. Please check this facts, since the information on probes and primers is essential to readers in order to reproduce the experiment.
-Lines 133-135. Can you explain what you mean by “Standard strains of caprine arthritis encephalitis and plasmids were synthesized by Huajin Biological Company”?. Do you mean the complete virus is re-constructed? Or just the sequence of the target gene?
-Line 139. Do you mean by using a tissue grinding bead beater? A bead is less than 1 mm width…
-Line 140. A nucleic acid extraction kit, not the nucleic acid extraction kit (there are dozens in the market)
-Line 146: contain
-Point 2.6. Again, please clarify the content of the standard.
-Line 162: This or these?
-Line 168: the phrase “The Ct value was determined by the reaction” must be explained: the reaction itself does not determine de Ct value.
-Table 3. I suggest writing “goat blood” and “sheep blood” in the right places in the table.
-Point 2-10. Unless you specify a procedure to analyze the genetic evolution of the virus, what you are presenting is a phylogenetic tree, very probably by performing Neighbor-Joining analysis. Clustal V result correspond to your figure 1 (alignment of the sequences), regardless the software you use for this. The exact definition of the methods and the procedures is critical in a scientific paper. Please re-write this point totally and make sure the procedures are correctly described. Also, specify the use of bootstrapping procedure and the support for the branches.
-Point 3.2. What did you clone in the plasmid?
-Point 3.6. The three samples detected by conventional RT PCR were among the six detected by using Taqman? If so, please state it. If this was not the case, a discussion on this fact will ne be necessary…
-Point 3.6 and 3.7. The 780 samples are said to be of sheep origin in 3.6. Then. In 3.7, 634 are from sheep and 146 from goats. Something looks wrong…
-Point 3.7 and 3.8. Before going on, please make sure you state the exact number of positive isolates. At point 3.7, you mention six positive samples. At point 3.8 you talk about one isolate from Shanghai and two from Jiangsu (I guess the three red point correspond to these three sequences, even if the figure 4 footnote does not explain it. Something is happening!
-Your sequences must be available at Genbank. So that you must submit them and provide the accession codes in your report. This is the only way to re-calculate your NJ tree, for instance.
The manuscript CAN NOT BE ACCEPTED before you deposit the sequences in the Genbank and provide the corresponding accession number in your text.
-In all, lines 248-256 deserve a more precise description. In the three, JS-2023-1 and JS-2023-2 appear as the most similar to each other. The Genbank Sequences similar to JS-2023-1 and JS-2023-2 and to SH-2023 have different accession numbers, so that they are different sequences, almost certainly obtained from different isolates. I strongly recommend considering the writing of this paragraph and asking for the help of English native professionals to produce a comprehensible and clear discussion.
-Line 295. The style of the references must be maintained, using numbers in brackets along the complete manuscript. The same occurs along this long paragraph, since the authors names are written and the reference style is not respected.
-Line 306: a 2011 Taqman assay obtained a much higher prevalence that the one presented in this manuscript. A discussion about these discordant results is necessary. Also, if the 2011 methods was efficient, you should discuss the need for a new one based up in the same methodology.
-Line 340. Before limiting the transmission of the virus limited to the Chinese regions, I strongly suggest making a Blastn search in Genbank to detect all over the world the sequences similar to the ones obtained in the present work. I can not do this search since your sequences are neither identified in Genbank or made available as supplementary material.
Comments on the Quality of English LanguageThe manuscript requires extensive correction before being acceptable- Also the Genbank accession numbers for the sequences must be available: OTHERWISE, THE MANUSCRIPT CAN NOT BE ACCEPTED.
Besides, at particular points, I have marked these comments:
-Line 21: The phrase “ This method demonstrated excellent 22 sensitivity, specificity, and reproducibility “ needs a demonstration, even in the abstract. For a given test Sensitivity is usually defined as the probability of a positive test result, when an individual is truly positive, while specificity is the probability of a negative result when an individual is a real negative. Both must be compared somehow with the diagnosis obtained in a different way. Lines 29-31 later explain that sensitivity is tested “in vitro” by using known concentrations of a plasmid carrying the GAG gene. Please clarify this “in vitro” validation for both sensitivity and specificity.
-Line 33. The term “genomes” usually refers to complete genomes. For a single gene (like GAG), it is more adequate to write “the sequence of the GAG gene” (for instance)
-Line 41 and successive: I strongly recommend dividing the first very long paragraph in the Introduction into, at least, three shorter paragraphs: very different ideas are exposed, and they surely deserve
-Line 49: why using the words “containing” and “including”?
-Line 53: This gene or this protein? Or the RNA transcribed from this gene? Please write a more clear sentence.
-Lines 59-61. This sentence requires a reference: in case the virus has been demonstrated to have reached China in this way, there must be a publication somewhere reporting this fact.
-Lines 61 and 62: again, a reference is needed to support the present low prevalence.
-Line 66: please re-write the phrase “The main prevalent strains in China are all subtype B1” and “The routes of CAEV transmission include: horizontal and vertical transmission”
-Line 70: do you mean transplacental transmission?
-Line 73: I suggest “...; goats are the most …”
-Line 76. I suggest “affected sheep”
-Line 77. Please re-write this phrase.
-Line 89-92. Please re-write this phrase on the different procedures. The present form is a bit confusing.
-Line 99: Please consider a different word for the “level” of the operator (for instance “highly skilled operators”?)
-Line 108: the number of brain samples and of blood samples is required. Also, goat samples are obtained in sheep farms in the line 110: is this correct?
-Line 119-121: the word compared appears twice. Please re-write this phrase.
-Line 123-124: either you explain how the primers were evaluated or you eliminate this phrase on the evaluation.
-Table 1: Usually the numbering is used to explain the position of the primers. This Table seems to explain the size of the product. Besides, the term (V = A/G/C). is not necessary since no V is visible in the sequences. Please check this facts, since the information on probes and primers is essential to readers in order to reproduce the experiment.
-Lines 133-135. Can you explain what you mean by “Standard strains of caprine arthritis encephalitis and plasmids were synthesized by Huajin Biological Company”?. Do you mean the complete virus is re-constructed? Or just the sequence of the target gene?
-Line 139. Do you mean by using a tissue grinding bead beater? A bead is less than 1 mm width…
-Line 140. A nucleic acid extraction kit, not the nucleic acid extraction kit (there are dozens in the market)
-Line 146: contain
-Point 2.6. Again, please clarify the content of the standard.
-Line 162: This or these?
-Line 168: the phrase “The Ct value was determined by the reaction” must be explained: the reaction itself does not determine de Ct value.
-Table 3. I suggest writing “goat blood” and “sheep blood” in the right places in the table.
-Point 2-10. Unless you specify a procedure to analyze the genetic evolution of the virus, what you are presenting is a phylogenetic tree, very probably by performing Neighbor-Joining analysis. Clustal V result correspond to your figure 1 (alignment of the sequences), regardless the software you use for this. The exact definition of the methods and the procedures is critical in a scientific paper. Please re-write this point totally and make sure the procedures are correctly described. Also, specify the use of bootstrapping procedure and the support for the branches.
-Point 3.2. What did you clone in the plasmid?
-Point 3.6. The three samples detected by conventional RT PCR were among the six detected by using Taqman? If so, please state it. If this was not the case, a discussion on this fact will ne be necessary…
-Point 3.6 and 3.7. The 780 samples are said to be of sheep origin in 3.6. Then. In 3.7, 634 are from sheep and 146 from goats. Something looks wrong…
-Point 3.7 and 3.8. Before going on, please make sure you state the exact number of positive isolates. At point 3.7, you mention six positive samples. At point 3.8 you talk about one isolate from Shanghai and two from Jiangsu (I guess the three red point correspond to these three sequences, even if the figure 4 footnote does not explain it. Something is happening!
-Your sequences must be available at Genbank. So that you must submit them and provide the accession codes in your report. This is the only way to re-calculate your NJ tree, for instance.
The manuscript CAN NOT BE ACCEPTED before you deposit the sequences in the Genbank and provide the corresponding accession number in your text.
-In all, lines 248-256 deserve a more precise description. In the three, JS-2023-1 and JS-2023-2 appear as the most similar to each other. The Genbank Sequences similar to JS-2023-1 and JS-2023-2 and to SH-2023 have different accession numbers, so that they are different sequences, almost certainly obtained from different isolates. I strongly recommend considering the writing of this paragraph and asking for the help of English native professionals to produce a comprehensible and clear discussion.
-Line 295. The style of the references must be maintained, using numbers in brackets along the complete manuscript. The same occurs along this long paragraph, since the authors names are written and the reference style is not respected.
-Line 306: a 2011 Taqman assay obtained a much higher prevalence that the one presented in this manuscript. A discussion about these discordant results is necessary. Also, if the 2011 methods was efficient, you should discuss the need for a new one based up in the same methodology.
-Line 340. Before limiting the transmission of the virus limited to the Chinese regions, I strongly suggest making a Blastn search in Genbank to detect all over the world the sequences similar to the ones obtained in the present work. I can not do this search since your sequences are neither identified in Genbank or made available as supplementary material.
Reviewer 2 Report
Comments and Suggestions for Authors
Manuscript ID: vetsci-2869158
Title: Detection and Phylogenetic Analysis of Caprine Arthritis-Encephalitis Virus by TaqMan-based qPCR in Eastern China
CAEV in goats and MVV in sheep are persistent viral infections caused by closely related lentiviruses, often grouped together as small ruminant lentiviruses (SRLVs). However, there is cross-species transmission between sheep and goats, with no clear evidence that one virus originated from the other. Most infected sheep and goats do not show clinical disease, but remain persistently infected and are capable of transmitting the virus. There is no commercially available vaccine against these diseases, which are on List B of the WOAH. Therefore, the main method of control and prevention is early detection and surveillance program (biosecurity management). Nevertheless, detection of SS-RNA virus is not easy. The gold standard method is virus isolation followed by molecular confirmation, but it is time consuming.
Real-time or quantitative PCR techniques are used in some laboratories and these tests, in addition to determining the infection status, also quantify the amount of MV or CAE provirus in an animal. In this study, a set of primers and tags was also developed for qPCR detection and surveillance of the virus in some Chinese farms.
The manuscript needs to improve and reorganize the writing structure with more detailed information about the methods used and well discussion of the results. Otherwise, it cannot be accepted for publication.
Abstract
Must be rewritten according to the standard structure: Introduction, Methods, Results, and Conclusion. Surveillance details such as number, location, data, type of samples tissue or whether are from clinicals diseased goat sheep or are subclinical samples. Also, information about analyzed sequences such as lengths should be added.
Introduction
Line 51. If Gag is the most conserved gene how used for phylogenic analysis? There is another study using this gene for detection and phylogeny? This should be clarified.
Line 64- Previous studies?
Line 73- CAEV is the main pathogen of the goat, but it can also be detected in the sheep mainly in a subclinical form?
Line 81. Control of the infection now is relying on biosecurity management
Line 97. The gold standard method is virus isolation followed by molecular confirmation
-Why only CAEV and not MVV, which is more prevalent in sheep?
-Previous work on qPCR detection of the virus should be discussed with the novelty and aim of your work. Probably move the first 2 paragraphs of discussion here.
Materials and Methods
-Surveillance and samples details such as farm location and number of samples from each one, date (month, year), type of samples tissue or whether are from clinicals diseased goat sheep or are subclinical samples should be provided in new tables. Moreover, the method of collection should also be added.
Provide the clinical sign of sampled animals
- why strain Shanxi has used for primer design?
- why instead of 58 ℃ of primer temperature used 53 ℃ for the TaqMan RT-qPCR assay?
- add the position of the amplifying sequence for each primer 271-290 and 338-317 and Probe.
-Method of sequences analysis and phylogenic tree should provide in detail. It is not clear which sequence size of gag compared? All of this information should be added to M and M and the legend of relevant figs.
Results
-Move Tables 2 and 3 to the Results section with more details.
-There are also sequence alignments for differential distinguish if yes should be added to Supplementary.
-This study needs a reorganization of the way the data are presented. It is not clear which samples are positive or negative. Provide details on the specificity of the assay for sheep and goat for clinical and subclinical samples or for blood and brain samples. All should be compared.
-Provide clinical signs of the positive animals.
Discussion
- The first 3 paragraphs are an introduction and are not required.
-Compare the sensitivity of your assay with previous work.
-Compare the sequence data with previous work Which position was subtitled and why? Why gag used?
Comments on the Quality of English Language
A minor improvement is required.
Reviewer 3 Report
Comments and Suggestions for Authors
Dear Author(s),
the topic about the detection and phylogenetic analysis of CAEV,addressed by you, is interesting as well as the adopted approach of the novel RT-qPCR assay, that has been proven efficient and robust.
Anyway there are various observations about your work.
In the Abstract the last sentence has to be reformulated, because it is not clear.
In Materials and Methods section the term “more” at line 109 has to be removed.
Has the in silico specificity by BLAST been verified during the primer and probe design, before performing the analyses?
Please, moved and integrated the paragraph 2.3 to the paragraph 2.7, because more pertinent.
In the title and, mainly, in the subtitles of all the paragraphs, please remove “TaqMan real-time quantitative polymerase chain reaction” in full. The abbreviation RT-qPCR is enough.
In the paragraph 2.5 please indicate the final concentration (in nM) of the primers.
Line 162-163: please, reformulate the sentence because it is unclear.
Line 169: please substitute the term reaction with amplification.
Please, better specify which and how many samples were used for the comparative analysis and in general for assay optimization.
Information and primer sequences used in conventional PCR are lacking or incomplete.
Table 2: please, review some CV% values.
Figure 2: enlarge, improving the quality, and better position the three images/figures. Please, do the same thing for Figure 3.
In the Discussion section (lines 260-285) there are redundant information, already reported in Introduction and more suitable for this section. Please, revised it.
The nested PCR, largely applied for SRLVs (CAEV/MVV) detection has not ever mentioned by you in your work.
In general, the bibliography is poor. Some relevant, similar or recent references about the topic are missing, and they should be added, such as:
· L'Homme Y, et al. Molecular characterization and phylogenetic analysis of small ruminant lentiviruses isolated from Canadian sheep and goats. Virol J. 2011;8:271. Published 2011 Jun 3. doi:10.1186/1743-422X-8-271
· De Regge N, Cay B. Development, validation and evaluation of added diagnostic value of a q(RT)-PCR for the detection of genotype A strains of small ruminant lentiviruses. J Virol Methods. 2013;194(1-2):250-257. doi:10.1016/j.jviromet.2013.09.001
· Marinho RC, et al. Duplex nested-PCR for detection of small ruminant lentiviruses. Braz J Microbiol. 2018;49 Suppl 1(Suppl 1):83-92. doi:10.1016/j.bjm.2018.04.013
· Arcangeli C, et al. First Survey of SNPs in TMEM154, TLR9, MYD88 and CCR5 Genes in Sheep Reared in Italy and Their Association with Resistance to SRLVs Infection. Viruses. 2021;13(7):1290. Published 2021 Jul 1. doi:10.3390/v13071290
And also other on phylo-genetic characterization of SRLVs.
Sincerely.

Comments on the Quality of English LanguageMinor editing of English language required
Round 2
Reviewer 1 Report
Comments and Suggestions for Authors
Please see the attached file

Comments on the Quality of English LanguageI have marked some phrases that should be improved
Author Response
1.-Line 36: You wrote:
Partial gag genome of the three positive samples were sequenced and compared with gag genome of other representative strains in genbank.
I suggest: “The Gag gene was partially sequenced in the three positive samples and compared with the sequences from other representative strains in GenBank”.
Also, the spelling GenBank is recommended along the manuscript.
Response:Thanks for the professional comment. We have revised the graph you referred in line 36 and maked it in red.
2.-Line 52-57 (and other positions): The names of the genes are not conserved along the text. For instance, you may read Gag, gag and GAG in this paragraph. The names must be consistent along the manuscript.
Response:Thanks for the professional comment. We have change all names to gag.
3.-Line 108. You wrote :
…and the highly skilled operators is required.
I suggest: …and highly skilled operators are required.
Response:Thanks for the professional comment. We have revised the graph you referred in line 108 and maked it in red.
4.-Lines 135-137. The words “…and compared” appear twice in the same phrase.
Response:We were sorry for this careless mistake.We have removed one of them.
5.-Line 148: tissue grinding bead brater or tissue grinding bead beater?
Response:We were sorry for this careless mistake. We have revised that in line 148 and marked it in red.
6.-Lines 172-176. Even in the new redaction, this paragraph is still a bit confusing.
According to my interpretation of the manuscript, three different procedures are
mentioned:
-The TaqMan RT-PCR procedure designed for the first time for the present
work.
-A RT-PCR (probably using SYBR green stain), that you usually denominate
“conventional RT PCR method”.
-A classical PCR using two primers and a standard thermal cycler. You call it
“conventional PCR method”. This provides the agarose gel shown in the figure 2B and the DNA for sequencing.
In all, this section should explain these three procedures in a simple way.
Response:Thanks for the professional comment. Actually, the “conventional RT-PCR” mentioned here means reverse transcription PCR not SYBR qPCR. We apologize for confusing the reviewers with these inappropriate expressions, and we have changed all conventional RT-PCR to conventional PCR.
7.-Lines 173.175. You wrote “In the conventional PCR method, we used Chinese
agricultural industry standard primers, the forward primer: AACTGGAAAGC AGTAGAC, the reverse primer:TACACTAGCTTGTTGCAC.” I suggest a simple redaction: “In the conventional PCR method, we used Chinese agricultural industry standard primers, (forward primer AACTGGAAAGCAG TAGAC; reverse primer TACACTAGCTTGTTGCAC).”
Response:Thanks for the professional comment. We have revised the graph you referred in line 173 and maked it in red.
8.-Figure 2c. I read “…Standara curve”. Please check the spelling of this word.
Response:We were sorry for this careless mistake. We have revised that in figure 2c.
9.-Line 280. Do you mean “We” or “we”?
Response:We were sorry for this careless mistake. We have revised that in line 280.

Reviewer 2 Report
Comments and Suggestions for Authors
The manuscript has improved somewhat, but not enough. The authors have not responded to the comments and have not added the information in the required manuscript and therefore cannot be accepted. CAEV is the disease of goat and in sheep it appears only in subclinical, please explain this clearly in your manuscript. As the manuscript describe they also detected only 3 subclinical sample that share close sequence identity. Another concern with this manuscript it is not clear that they used same primer of RT-qPCR for sequencing analysis which has 68 bp or another primer set! This is not acceptable because of short length and conserved sequences. As I checked the registered sequences have 571 bp! so the manuscript should contain this information with describing used methods.
The project has ethical approval to collect more than 700 clinical specimens? Add the code number. The method of sampling is not clear to check if there are any animal welfare or ethical issues, or problems with sampling protocols may be the reason for detection of only few number of positive samples.
Abstract
Add this to abstract. ‘These results suggest that the designed RT-qPCR assay can be used to detect subclinical CAEV in sheep, and the virus is still present in eastern China’.
-Method of sequences analysis and phylogenic tree should provide in detail. It is not clear which sequence size of gag compared? All of this information should be added to M and M and the legend of relevant figs.
Response: Thank you for your suggestion, we have demonstrated details in a new table (table 4).
Table 4 is not necessary, remove it. I think the authors didn't get what I asked, need to add details of phylogeny in caption of Fig.4. should add method of analysis and name of software, number of samples compared (how many are yours with name and acc no and compared with 28 sequences and length of your sequences. And indicate that is marked by red circle.
Why only CAEV and not MVV, which is more prevalent in sheep?
Response: MMV is more prevalent in sheep, but it's not popular in China. MMV is widely spread in Switzerland, Italy, Iceland, and elsewhere.
-This Should be added to the manuscript with more details.
-Discussion
There is major problem with discussion the author should follow scientific writing and structure. First explain your work, then one by one compare your results with previous works and explain the involved factors or reason of differences. Then the result should be clearly concluded with suggestions for future studies.
Comments on the Quality of English Language
Moderate editing of the English language is required.
Author Response
1.The manuscript has improved somewhat, but not enough. The authors have not responded to the comments and have not added the information in the required manuscript and therefore cannot be accepted. CAEV is the disease of goat and in sheep it appears only in subclinical, please explain this clearly in your manuscript. As the manuscript describe they also detected only 3 subclinical sample that share close sequence identity.
Response:Thank you very much for your professional comment. We have added this to introduction in line84.
“Clinical symptoms are observed in only 20 to 30 percent of affected goats; in sheep and other small ruminants, it only presents with subclinical symptoms.”
2.Another concern with this manuscript it is not clear that they used same primer of RT-qPCR for sequencing analysis which has 68 bp or another primer set! This is not acceptable because of short length and conserved sequences. As I checked the registered sequences have 571 bp! so the manuscript should contain this information with describing used methods.
Response:Thanks for the professional comment. The primer we used for sequencing is just the primer used in conventional PCR,it is Chinese agricultural industry standard primers, and the information of it we has been added in table 1.We have demonstrated this in line204.
“The primer set we used for sequencing is just the primer set we used in the convention-al PCR method, which was referred to in Section 2.6.”
3.The project has ethical approval to collect more than 700 clinical specimens? Add the code number. The method of sampling is not clear to check if there are any animal welfare or ethical issues, or problems with sampling protocols may be the reason for detection of only few number of positive samples.
Response:Thanks for the professional comment. We have ethical approval after main text including the code number.
“The study was conducted according to the animal ethics guidelines of China and approved by the Institutional Animal Care and Use Committee of Shanghai Veterinary Research Institute (IACUC No.: Shvri-gs-20220806-1, 6 August 2022).”
And we have tried our best to describe the method of sampling clearly in 2.1.
“Sheep samples (n=634; 630 blood samples and 4 brain samples) were collected between July and September 2021 from three sheep farms in Shanghai. In July 2022, we collected 146 goat blood samples from three goat farms in Jiangsu, China. Of all the clinical samples, only four sheep showed neurological symptoms, and the farm pro-vided the brains of these sheep. The blood samples were randomly collected from asymptomatic animals. All samples were delivered to the lab in Drikold and then stored at -80℃ until use. All experimental procedures involving animal were performed in adherence to the Guidelines for the Keeping and Use of Laboratory Animals, and approval for this study was obtained from the Animal Ethics Committee of the Shanghai Institute of Veterinary Medicine, Chinese Academy of Agricultural Sciences. The field study did not involve endangered or protected species. No specific permissions were required for the collection of samples because the samples were collected from public areas or nonprotected areas. We received approval from the farm owners for data sampling and the publication of the data.”
4.Abstract
Add this to abstract. ‘These results suggest that the designed RT-qPCR assay can be used to detect subclinical CAEV in sheep, and the virus is still present in eastern China’.
Response:Thanks for the professional comment. We have added this to abstract in line39.
5.-Method of sequences analysis and phylogenic tree should provide in detail. It is not clear which sequence size of gag compared? All of this information should be added to M and M and the legend of relevant figs.
Response: Thank you for your suggestion, we have demonstrated details in a new table (table 4).
Table 4 is not necessary, remove it. I think the authors didn't get what I asked, need to add details of phylogeny in caption of Fig.4. should add method of analysis and name of software, number of samples compared (how many are yours with name and acc no and compared with 28 sequences and length of your sequences. And indicate that is marked by red circle.
Response:Thanks for the professional comment. We have added this to Fig4.
“Phylogenetic tree of caprine arthritis encephalitis virus (CAEV) based on the partial gag gene (about 571bp). The three strains isolated in this study—JS-2023-1 (no.PP382771), JS-2023-2 (no.PP382772), and SH-2023 (no.PP382773)—are marked by red circles. The other 28 representative sequences were downloaded from GenBank and edited to obtain the corresponding segments using Editseq version 7.1.0 software (DNASTAR Inc., Madison, WI, United States; Lole et al., 1999) based on the se-quencing primers. Multiple sequence alignment was performed using the Clustal V method, and a phylogenetic analysis of the recombinants was performed using MegAlign version 7.1.0 software (DNASTAR Inc., Madison, WI, United States; Lole et al., 1999).”
6.Why only CAEV and not MVV, which is more prevalent in sheep?
Response: MMV is more prevalent in sheep, but it's not popular in China. MMV is widely spread in Switzerland, Italy, Iceland, and elsewhere.
-This Should be added to the manuscript with more details.
Response:Thanks for the professional comment. We have added this to introduction in line51.
“MVV is widely spread in Europe, including in Switzerland, Italy, Iceland, and else-where, and only rarely reported in China, with all of cases being located in the Sichuan and Xinjiang provinces; hence, this study focused on CAEV, which is more prevalent in China and covers a wider area [1,2]”
Reference
- Liu Diangui, Q.Z., Pu Fangde,Lin Jie,Li Jianjun. Serologic diagnosis of Maedi visna. Chinese Veterinary Science 1990, 12.
- Gong Chengyuan, Z.T., Hu Zeyuan,Lin Jie,Yue Min. Serologic diagnosis of Maedi visna in Xinjiang (preliminary report). Chinese Journal of Veterinary Medicine 1984, 12, 11-14.
7.-Discussion
There is major problem with discussion the author should follow scientific writing and structure. First explain your work, then one by one compare your results with previous works and explain the involved factors or reason of differences. Then the result should be clearly concluded with suggestions for future studies.
Response:Thanks for the professional comment. Based on the scientific writing and structure we have made major revision to the discussion.
First, we introduced our work.
“CAEV infections are prevalent in dairy goats all over the world, and a higher in-cidence of CAE has been noted in many industrialized nations. In this study, a high-sensitivity and high-specificity assay based on TaqMan real-time PCR amplifica-tion was developed for detecting CAEV. Using the method established in this study, we tested samples from flocks in eastern China (Shanghai and Jiangsu) and found rela-tively low CAEV positivity in eastern China (around 0.77%), but the virus has not been eliminated completely. This result was lower than expected, and we attribute this de-viation to the guidelines outlined in the document titled "Diagnostic Techniques for CAEV " (NY/T 3465-2019), issued in China in 2019. This document establishes the technical requirements for the clinical diagnosis of CAEV, protocols for the isolation and identification of pathogens, methods for detecting pathogenic nucleic acids, and guidelines for serological tests.”
Next, we compare this study with previous studies, describe the benefits and drawbacks of the various testing methods, and analyze why prevalence changes.
“In China, CAEV has been a concern for the caprine farming industry since the first domestic infection caused by CAEV in 1988. Epidemiological investigations of CAEV in eastern China trace back to 1995, at which time CAEV was not found to be circulating in goat flocks in that region. Since then, there have been no publicly published detec-tion reports on CAEV in eastern China. China put forward the "Eighth Five-Year Plan" national scientific and technological research project in the 1990s. The prevention and treatment of CAE was an important topic in this plan. During this period, Chinese sci-entists conducted an extensive examination covering a total of 11449 goats from 11 different breeds across 15 provinces in China. They employed clinical and serological methods, revealing varying degrees of CAEV prevalence in most provinces: Gansu (5.14%), Shaanxi (9.25%), Yunnan (0.52%), Guizhou (7.45%), Henan (1.08%), Shandong (6.16%), Hainan (4.18%), Liaoning (1.80%), Sichuan (0.42%), Xinjiang (0.18%), and Heilongjiang (0.11%) . ELISA methods are suitable for large-scale sample testing but are usually only used for antibodies in serum, and ELISAs require multiple steps to carry out, resulting in longer experimental times. Despite the ongoing efforts by scien-tists in the post-Eighth Five-Year Plan era, research on the prevention and control of CAE is still lacking, and the results concerning this topic are not fully conclusive. In 2012, a Chinese academic conducted tests on 308 clinical samples from Tianjin and Shaanxi, utilizing mononuclear peripheral blood cells from goat blood as target cells, resulting in a positive rate of 7.8%. In 2013, some Chinese researchers conducted an investigation on 308 goat samples collected between 1993 and 2011 in the Shanxi and Tianjin provinces of China. They employed a TaqMan RT-qPCR assay, revealing a positive rate of 7.8%. Compared to previous methods, the TaqMan RT-qPCR assay developed in this study demonstrates enhanced stability, reduced error rates in results, and heightened specificity compared to previously established RT-qPCR assays, par-ticularly in distinguishing CAEV from other prevalent infectious diseases in sheep. In 2018, a Chinese academic conducted tests on 165 goat samples from the Xinjiang re-gion, employing both molecular and serological assays, resulting in a positivity rate of 2.4%. In the same year, Chinese scientists tested 2,083 goat samples from 11 prov-inces in China, revealing a positivity rate of 0.38%. A comparison of the results of previous studies with the results of this study suggests that the prevalence of CAEV is higher in the western region of China than in the east and higher in the northern re-gion than in the south. We speculate that this may be because the practice of sheep farming is more developed in the northwestern region. Meanwhile, in East China re-gions like Shanghai and Jiangsu, where disease prevention and control measures, comprehensive management capabilities, and public safety regulations are more stringent, sheep farms enforce strict regulations on trade. This, we hypothesize, may have contributed to the lower positive rate observed in our study.”
Subsequently, we made an analysis of phylogenetic evolution and made some recommendations for prevention and control of CAEV.
“We tried to isolate the strains that tested positive, but it was difficult to do so. Hence, we resorted to sequencing to obtain the sequence of their gag gene, and com-pared the three isolates with the sequences already uploaded on NCBI to obtain a phylogenetic tree. From the tree, we inferred that subtype B has spread very widely, appearing in countries on many different continents, including China, the United States, Canada, Poland, Switzerland, France, Brazil, Italy, and Spain. Subtype B1 is still the most prevalent subtype in China. Compared to the northwestern and southwestern regions of China, the practice of sheep farming in eastern regions is less developed, and thus, sheep farmers in this region are more likely to neglect and underestimate the presence of infectious diseases in their flocks. As a chronic infectious disease, CAE is difficult for breeders to detect in the first instance, meaning that regular screening is necessary to prevent this disease from causing serious economic losses to the sheep farming industry. CAEV is more common in goats. However, there is evidence of the direct transmission of the SRLV subtype B1 from goats to sheep, and our study also reaffirms the validity of the transmission of subtype B1 in sheep. These results suggest it is best to raise goats and sheep separately and avoid mixing them”
Finally, we summarized the work and made expectations of the significance of it.
“CAEV eradication programs are hindered by late seroconversion, observed in goats, and by the absence of detectable antibodies in infected animals, which delays diagnosis and promotes the dissemination of the disease. Therefore, pathogenetic detection methods play an important role in CAEV eradication programs. The estab-lishment of the optimized TaqMan RT-qPCR assay provides a technical means for the clinical detection of CAEV. Not only can specific pathogens be rapidly detected in the early stages of epidemics and appropriate response strategies be formulated in a timely manner, but also the epidemiological and developmental trends of diseases can be monitored, and scientific research can be promoted. We hope that this assay can con-tribute to the sheep farming industry and provide long-lasting benefits.”

Reviewer 3 Report
Comments and Suggestions for Authors
The authors amended the manuscript accordingly to the suggestions and comments. Anyway, the comprehension of the text is not always linear, so limiting and losing the scientific soundness. I suggest to revise it also asking for support of a native English speaker.
Have been the sequences of CAEV strains already deposited in GeneBank?
The concentration of primer in nM is not correct, such as some information about conventional RT-PCR method is still lacking.
Comments on the Quality of English Language
Extensive editing of English language required.
Author Response
1.Have been the sequences of CAEV strains already deposited in GeneBank?
Response:Yes, we have submitted the sequences to GenBank and get acc number.
JS-2023-1 (no.PP382771)、JS-2023-2(no.PP382772) and SH-2023(no.PP382773)
2.The concentration of primer in nM is not correct, such as some information about conventional RT-PCR method is still lacking.
Response:Thanks for the professional comment. We have revised the unit of primers in line161 and marked it in red.
Actually, the “conventional RT-PCR” mentioned here means reverse transcription PCR not SYBR qPCR. We apologize for confusing the reviewers with these inappropriate expressions, and we have changed all conventional RT-PCR to conventional PCR.
- The authors amended the manuscript accordingly to the suggestions and comments. Anyway, the comprehension of the text is not always linear, so limiting and losing the scientific soundness. I suggest to revise it also asking for support of a native English speaker.
Response:Thanks for the professional comment. We have used English service of MDPI to improve the quality of English language in the text.

Round 3
Reviewer 2 Report
Comments and Suggestions for Authors
Manuscript is improved and proposed to be published by VS.
Comments on the Quality of English LanguageRequires minor English editing